# Meta-ticket: Finding optimal subnetworks for few-shot learning within randomly initialized neural networks

**Daiki Chijiwa**[1*]    **Shin'ya Yamaguchi**[1,2]    **Atsutoshi Kumagai**[1]    **Yasutoshi Ida**[1]

[1]NTT Computer and Data Science Laboratories, NTT Corporation
[2]Kyoto University

## Abstract

Few-shot learning for neural networks (NNs) is an important problem that aims to train NNs with a few data. The main challenge is how to avoid overfitting since over-parameterized NNs can easily overfit to such small dataset. Previous work (e.g. MAML by Finn et al. 2017) tackles this challenge by meta-learning, which learns how to learn from a few data by using various tasks. On the other hand, one conventional approach to avoid overfitting is restricting hypothesis spaces by endowing sparse NN structures like convolution layers in computer vision. However, although such manually-designed sparse structures are sample-efficient for sufficiently large datasets, they are still insufficient for few-shot learning. Then the following questions naturally arise: (1) Can we find sparse structures effective for few-shot learning by meta-learning? (2) What benefits will it bring in terms of meta-generalization? In this work, we propose a novel meta-learning approach, called **Meta-ticket**, to find optimal sparse subnetworks for few-shot learning within randomly initialized NNs. We empirically validated that Meta-ticket successfully discover sparse subnetworks that can learn specialized features for each given task. Due to this task-wise adaptation ability, Meta-ticket achieves superior meta-generalization compared to MAML-based methods especially with large NNs.

## 1   Introduction

Recent neural networks (NNs) have achieved remarkable performance for solving complex recognition tasks such as image recognition and natural language understanding [20, 58]. These successful performance are mainly due to carefully designed NN structures (inductive biases) and a large amount of training data. However, it still takes significant cost for collecting such a large amount of data. Thus, **few-shot learning** has been an important problem that aims to enable NNs to learn from a few samples or experiences like humans can do. Since NNs have strong memorization capacity [67], the main challenge of few-shot learning is how to avoid overfitting to a small number of data.

**Meta-learning** is one of the promising approaches for few-shot learning, which can automatically learn how to learn without overfitting, by using a bunch of tasks from a so-called meta-training dataset. Meta-learning consists of two optimizations: adapting to a given task (inner optimization) and improving the inner optimization for each task (meta-optimization). Gradient-based meta-learning [14] is an approach modeling the inner optimization as gradient descent steps, which leads to sample-efficient meta-learning compared to black-box models like recurrent NNs [22]. Model-Agnostic Meta-Learning (MAML, proposed by Finn et al. [13]) is a representative method of the gradient-based meta-learning. It optimizes initial parameters in a given NN with stochastic gradient descent (SGD) so that it can adapt to a novel task by gradient descent starting from the meta-learned initial parameters. However, Raghu et al. [50] pointed out that, in classification tasks with a NN,

---

*Corresponding author: `daiki.chijiwa@ntt.com`

36th Conference on Neural Information Processing Systems (NeurIPS 2022).

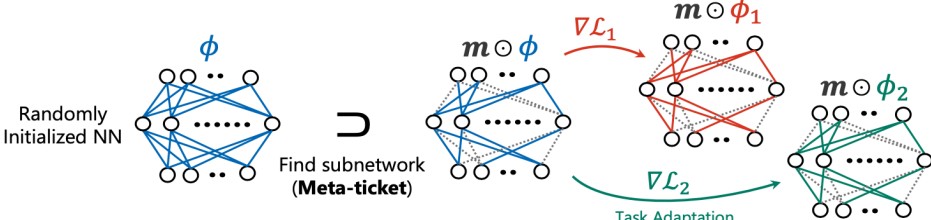

Figure 1: For a randomly initialized neural network $f_\phi(x)$, instead of optimizing the initial parameters $\phi$, Meta-ticket finds an optimal subnetwork $f_{\mathbf{m}\odot\phi}(x)$ by meta-optimizing binary parameters $\mathbf{m}$ so that the subnetwork can successfully adapt to tasks with a few samples.

MAML tends to fix its feature extractor and learn only its classification layer (**feature reuse**) rather than learning both the feature extractor and the classification layer (**rapid learning**) during inner optimization. In other words, *MAML directly encodes information of the meta-training dataset into the meta-learned initial parameters in the feature extractor*, which results in suboptimal solutions that cannot sufficiently adapt to tasks in unknown domains [44].

On the other hand, another conventional approach to avoid overfitting is to limit hypothesis spaces for learning. In the case of NNs, this is corresponding to designing domain-specific NN structures. The structure of convolutional neural networks (CNNs) is one of the most successful examples. Neyshabur [42] pointed out that the superior generalization capacity of CNNs is mainly due to their sparse structures. However, although such hand-designed sparse NN structures are sample-efficient with sufficiently large dataset, they are still insufficient for few-shot learning [59].

In this paper, we focus on sparse NN structures for few-shot learning rather than the initial parameters like MAML. Our research questions are: (1) Can we automatically find sparse NN structures that are effective for few-shot learning by meta-learning? (2) What are the strengths and weaknesses of the meta-learned NN structures compared to the meta-learned initial parameters of MAML? To answer these questions, we exploit techniques to identify sparse subnetworks within a randomly initialized NN from the literature of strong lottery ticket hypothesis [51]. Based on them, we formulate a novel gradient-based meta-learning framework called **Meta-ticket** in a parallel way to MAML, which finds an optimal subnetwork for few-shot learning within a given randomly initialized NN (Figure 1). Specifically, Meta-ticket meta-optimizes the subnetwork structures instead of the initial parameters. Our findings are summarized as follows:

- We validated that Meta-ticket can discover sparse NN structures achieving comparable few-shot performance to MAML on few-shot image classification benchmarks (Section 4), even though Meta-ticket never directly meta-optimize the randomly initialized parameters in NNs. Although Meta-ticket often degrades with small NNs on easy tasks, it outperforms MAML when using larger NNs on more difficult tasks. This indicates that the meta-learned NN structures enable the scaled-up NNs to exploit their increased capacity with less overfitting in few-shot learning.

- Moreover, we found that *Meta-ticket prefers rapid learning rather than feature reuse* in contrast to MAML. This remarkable property may be emerged from the different meta-optimization, suggested by rough theoretical arguments in Section 3.2. As a result, Meta-ticket outperforms MAML-based methods by a large margin (up to more than 15% in accuracy) in cross-domain evaluations (Section 4.1.3, 4.2).

- Also we analyzed the effect of sparse NN structures themselves for few-shot learning. To remove the effect of initial parameter values, we consider the setting that parameters are initialized with the same constant. Even in such an extreme setting, Meta-ticket achieves over 70% accuracy while random subnetworks only do nearly 30% (Section 4.1.2).

## 2 Background

### 2.1 Meta-learning for few-shot learning

Let $X$ be an input space and $Y$ an output space. A distribution $\mathcal{T}$ over $X \times Y$ is called a *task*. In a standard learning problem, given a task $\mathcal{T}$, we consider a model $f_\phi(x) : X \to Y$ parameterized

by $\phi \in \mathbb{R}^n$ and optimize $\phi$ to reduce the expectation value of a loss function $\mathcal{L} : Y \times Y \to \mathbb{R}$. For simplicity, we denote the mean value of the loss function over a dataset $\mathcal{D} \subset X \times Y$ as $\mathcal{L}(f_\phi; \mathcal{D}) := \frac{1}{|\mathcal{D}|} \sum_{(x,y) \in \mathcal{D}} \mathcal{L}(f_\phi(x), y)$.

In meta-learning, we consider a distribution $p(\mathcal{T})$ over the set of tasks $\mathcal{T}$. Meta-learning consists of two optimization stages: adapting to a given task $\mathcal{T}$ (*inner optimization*) and improving the inner optimization for each task (*meta-optimization*). To model the inner optimization, we consider a meta-model $F_\theta(\mathcal{D})$ parameterized by a meta-parameter $\theta \in \mathbb{R}^N$, where $\mathcal{D}$ is a training data sampled i.i.d. from the task $\mathcal{T}$. Given $k \in \mathbb{N}$, we consider the setting of $k$-*shot learning* where the training data $\mathcal{D}$ consists of $k$ samples, and denote a task with $k$ samples over $(X \times Y)^k$ by $\mathcal{T}^k$. Then a meta-model for $k$-shot learning is given in the form of $F_\theta : (X \times Y)^k \to \mathbb{R}^n$ with a meta-parameter $\theta \in \mathbb{R}^N$. The aim of meta-learning for $k$-shot learning is to optimize the meta-parameter $\theta$ as follows:

$$\min_{\theta \in \mathbb{R}^N} \mathbb{E}_{\mathcal{T} \sim p(\mathcal{T})} \mathbb{E}_{\mathcal{D}^{\mathrm{tr}}, \mathcal{D}^{\mathrm{val}} \sim \mathcal{T}^k} \left[ \mathcal{L}(f_{\phi'}; \mathcal{D}^{\mathrm{val}}) \right], \quad \text{where } \phi' = F_\theta(\mathcal{D}^{\mathrm{tr}}). \tag{1}$$

The meta-parameter $\theta$ is optimized to minimize the expected validation loss $\mathcal{L}(f_{\phi'}; \mathcal{D}^{\mathrm{val}})$ with $\phi'$ adapted to a training data $\mathcal{D}^{\mathrm{tr}}$ by the meta-model $F_\theta$.

## 2.2 Model-agnostic meta-learning (MAML)

MAML [13] is a representative method of gradient-based meta-learning [14] which optimizes hyperparameters appearing in gradient descent procedure for $k$-shot learning. Specifically, MAML learns an initial parameter $\phi_0 \in \mathbb{R}^n$ for a model $f_\phi(x)$. Here we focus on the case $f_\phi(x)$ is an $L$-layered neural network parametrized by $\phi = (\phi^l)_{l=1 \cdots, L} \in \oplus_{l=1,\cdots,L} \mathbb{R}^{n_l}$, where $n_l$ is the number of parameters in the $l$-th layer. The meta-model of MAML (with $S$ gradient steps) for a neural network $f_\phi(x)$ is described as follows:

$$F_{\phi_0}^{\mathrm{MAML}}(\mathcal{D}^{\mathrm{tr}}) = \phi_{S-1}$$
$$= \left( \phi_0^l - \alpha^l \sum_{i=0}^{S-1} \nabla_{\phi_i^l} \mathcal{L}(f_{\phi_i}; \mathcal{D}^{\mathrm{tr}}) \right)_{1 \le l \le L}, \tag{2}$$

where we set $\phi_i := (\phi_i^l)_{1 \le l \le L}$ as the collection of parameters at $i$-th step, and $\phi_i^l := \phi_{i-1}^l - \alpha^l \nabla_{\phi_{i-1}^l} \mathcal{L}(f_{\phi_{i-1}}; \mathcal{D}^{\mathrm{tr}})$ as an $i$-th gradient step for $l$-th layer. $\alpha^l \in \mathbb{R}_{\ge 0}$ is called an inner learning rate for $l$-th layer, and the gradients $\nabla_{\phi=\phi_i} \mathcal{L}(f_\phi; \mathcal{D}^{\mathrm{tr}})$ in the inner optimization are called *task gradients*.

## 2.3 Rapid learning vs. feature reuse

Raghu et al. [50] investigated what makes the MAML algorithm effective in few-shot classification with neural networks. Let $f_\phi(x) = h_{\phi_L} \circ g_\Phi(x)$ be an $L$-layered neural network, where we denote an $(L-1)$-layered neural network $g_\Phi(x)$ (*feature extractor*) and a linear classifier $h_{\phi_L}$ (*output layer*). Under these settings, they considered two phenomenon: **rapid learning** and **feature reuse**. Rapid learning means that the neural network $f_\phi(x)$ adapt to each task with large changes in the overall parameters $\phi$. Feature reuse means the feature extractor $g_\Phi(x)$ meta-learns extracting reusable features for all tasks and is barely changed during the inner optimization. Thus only the output layer $h_\Phi(x)$ adapts to each task in the feature reuse case. The question posed by Raghu et al. [50] is as follows: Which of rapid learning and feature reuse is dominant in meta-models trained with MAML?

They empirically showed that the task gradients of the feature extractor in MAML approximately vanish, and thus the feature reuse is dominant in MAML, on few-shot classification benchmarks. Also they introduced a variant of MAML, called ANIL, that lets all parameters of the feature extractor $g_\Phi(x)$ be frozen in inner optimization (i.e. setting the inner learning rates $\alpha^l = 0$ for $1 \le l \le L-1$). Since ANIL fully satisfies feature reuse by definition, it performs similarly to MAML.

On the other hand, Oh et al. [44] investigated the rapid learning aspects of MAML. They proposed another variant of MAML, called BOIL, that let the parameter of the output layer be frozen in inner optimization (i.e. $\alpha^L = 0$). In other words, contrary to ANIL, BOIL forces the parameters in the feature extractor to largely change for each task since the parameter of the output layer cannot adapt anymore during inner optimization. Although this formulation seems counterintuitive, they showed that BOIL actually works well. Moreover, BOIL outperforms MAML and ANIL on cross-domain

adaptation benchmarks, i.e. evaluating meta-learners with datasets that differ from the one used for meta-learning. The result indicates that the feature reuse in MAML may be suboptimal from the viewpoint of meta-generalization.

# 3 Methods

In this section, we propose a novel gradient-based meta-learning method called **Meta-ticket** (Algorithm 1). The aim of Meta-ticket is to meta-learn an optimal sparse NN structure for $k$-shot learning, within a randomly initialized NN. The algorithm can be formulated in parallel to MAML as we will see in Section 3.1. In contrast to MAML, which directly encodes information of a meta-training dataset into the NN parameters, Meta-ticket instead meta-learns which connections between neurons are useful for learning new tasks. Due to this difference in meta-optimization, it is shown that Meta-ticket prefers rapid-learning rather than feature reuse by analyzing task gradients in Section 3.2, which leads to better cross-domain adaptation as demonstrated in Section 4.

## 3.1 Algorithm: Meta-ticket

Let $f_\phi(x)$ be an $L$-layered neural network parametrized by $\phi = (\phi^l)_{1 \le l \le L} \in \oplus_{1 \le l \le L} \mathbb{R}^{n_l} = \mathbb{R}^n$, where $n_l$ is the number of parameters in the $l$-th layer and $n := \sum_{1 \le l \le L} n_l$. We define a sparse structure for $f_\phi(x)$ as a binary mask $\mathbf{m} = (\mathbf{m}^l)_{1 \le l \le L} \in \oplus_{1 \le l \le L} \{0, 1\}^{n_l} = \{0, 1\}^n$ whose dimension is same as $\phi$. The corresponding sparse subnetwork is $f_{\mathbf{m} \odot \phi}(x)$ where $\odot$ means an element-wise product in $\mathbb{R}^n$. The meta-model $F_{\mathbf{m}}^{\mathrm{MT}}(\mathcal{D}^{\mathrm{tr}})$ for Meta-ticket (with $S$ gradient steps) is described as follows:

$$F_{\mathbf{m}}^{\mathrm{MT}}(\mathcal{D}^{\mathrm{tr}}) = \mathbf{m} \odot \phi_{S-1}$$

$$= \mathbf{m} \odot \left( \phi_0^l - \alpha^l \sum_{i=0}^{S-1} \nabla_{\phi_i^l} \mathcal{L}(f_{\mathbf{m} \odot \phi_i}; \mathcal{D}^{\mathrm{tr}}) \right)_{1 \le l \le L}, \tag{3}$$

where $\alpha^l \in \mathbb{R}$ is an $l$-th inner learning rate, $\phi_i^l := \phi_{i-1}^l - \alpha^l \nabla_{\phi_{i-1}^l} \mathcal{L}(f_{\mathbf{m} \odot \phi_{i-1}}; \mathcal{D}^{\mathrm{tr}})$ is an $i$-th gradient step, and $\phi_i := (\phi_i^l)_{1 \le l \le L}$. The differences in Eq. (3) from Eq. (2) of MAML are the parts involving the mask $\mathbf{m}$. In contrast to MAML optimizing the initial parameter $\phi_0$, Meta-ticket optimizes only the binary mask $\mathbf{m}$ and keeps $\phi_0$ being a fixed randomly initialized parameter.

The whole algorithm of Meta-ticket is described in Algorithm 1, which involves two functions `CalculateMask()` and `UpdateScore()` to deal with the meta-parameter $\mathbf{m}$. Since $\mathbf{m}$ is a discrete variable, we cannot directly apply the standard NN optimization methods to it. Instead, we optimize a latent continuous parameter $\mathbf{s} = (\mathbf{s}^l)_{1 \le l \le L} \in \mathbb{R}^n$ corresponding to $\mathbf{m}$, called a *score parameter*, inspired by Ramanujan et al. [51].

More specifically, we introduce a hyperparameter $p_{\mathrm{init}} \in [0, 1]$ called an *initial sparsity*, which determines the ratio of how many components in $\mathbf{m}$ become zero at the beginning of the algorithm. Also we randomly initialize the score parameter $\mathbf{s}$. Using the initial sparsity $p_{\mathrm{init}}$ and the initialized score $\mathbf{s}$, for each layer $l \in \{1, \cdots, L\}$, we set the *layer-wise score threshold* $\sigma_l \in \mathbb{R}$ as the $\lfloor p_{init} n_l \rfloor$-th smallest score $s_{i*}$ in the score parameters $\{s_i : 1 \le i \le n_l\}$ in the $l$-th layer, before the algorithm starts. During the algorithm, using the fixed thresholds, `CalculateMask(s)` returns the layer-wise masks $(\mathbf{m}^l)_{1 \le l \le L}$ where each component $m_i^l$ of $\mathbf{m}^l$ is given by

$$m_i^l = \begin{cases} 0 & (s_i^l \le \sigma_l), \\ 1 & (s_i^l \ge \sigma_l). \end{cases} \tag{4}$$

Although $\mathbf{m}$ is a non-differentiable function in the latent variable $\mathbf{s}$, we can optimize $\mathbf{s}$ by straight-through estimator [7] as previous literature [51]. The optimization step is given by `UpdateScore(s, $\mathcal{L}_{\mathbf{m}}$) := $\mathbf{s} - \beta \nabla_{\mathbf{m}} \mathcal{L}_{\mathbf{m}}$ + (momentum)`, i.e. gradient descent with the gradient for $\mathbf{m}$ with respect to the meta-objective $\mathcal{L}_{\mathbf{m}}$ defined by Eq. (1), (3), and an outer learning rate $\beta$.

## 3.2 Analysis on task gradients

The meta-models learned with Meta-ticket could have different properties from the ones with MAML since their meta-optimization targets are different. One is a discrete variable $\mathbf{m}$ (Meta-ticket) and

**Algorithm 1** Meta-ticket    (The colored codes are unique to this algorithm.)

**Require:** $\alpha, \beta \in \mathbb{R}_{\geq 0}, p_{\text{init}} \in [0, 1]$        ▷ $\alpha, \beta$: inner/outer learning rates, $p_{\text{init}}$: an initial sparsity.
**Require:** $\phi_0 \in \mathbb{R}^n, \mathbf{s} \in \mathbb{R}^n$                                ▷ Randomly initialized parameters.
**Require:** $(\sigma_l)_{1 \leq l \leq L} \in \mathbb{R}^L$        ▷ Score thresholds determined by $p_{\text{init}}$ and the initial score $\mathbf{s}$.
 1: $\mathbf{m} \leftarrow \texttt{CalculateMask(s)}$           ▷ Initialize the binary mask $\mathbf{m}$ using the initial score $\mathbf{s}$
 2: **while s** is not converged **do**
 3:      Sample $B$ tasks $\mathcal{T}_1, \cdots, \mathcal{T}_B \sim p(\mathcal{T})$
 4:      **while** $b \leftarrow 1, \cdots, B$ **do**
 5:          Sample datasets $\mathcal{D}_b^{\text{tr}}, \mathcal{D}_b^{\text{val}} \sim \mathcal{T}$ for $k$-shot learning of the task $\mathcal{T}_b$
 6:          $\phi_b' \leftarrow F_{\mathbf{m}}^{\text{MT}}(\mathcal{D}_b^{\text{tr}})$
 7:      **end while**
 8:      Calculate the meta-loss $\mathcal{L}_{\mathbf{m}} = \frac{1}{B} \sum_{b=1}^B \mathcal{L}\big(f_{\phi_b'}(x); \mathcal{D}_b^{\text{val}}\big)$
 9:      $\mathbf{s} \leftarrow \texttt{UpdateScore}(\mathbf{s}, \mathcal{L}_{\mathbf{m}})$
10:      $\mathbf{m} \leftarrow \texttt{CalculateMask(s)}$
11: **end while**
12: return $\mathbf{m}$

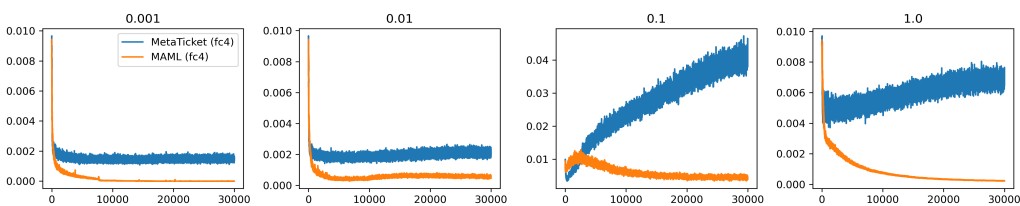

Figure 2: We plotted the mean absolute values of task gradients for the last layer of the feature extractor in a 5-layered MLP (meta-trained on CIFAR-FS as in Section 4.1.3) with varying inner learning rates.

another is a continuous variable $\phi_0$ (MAML). As discussed in Section 2.3, feature reuse is dominant in the task gradients in MAML, and it may be harmful for meta-generalization capacity. Then a natural question arises: What about the task gradients in Meta-ticket? Here we compare the dynamics of task gradients during meta-training between MAML and Meta-ticket, both empirically and theoretically.

First of all, we performed experiments to see how task gradients of the feature extractor change during meta-training, for different inner learning rate setting (Figure 2). We find that the task gradients in both methods behave similarly in very early phase of meta-training. However, in the rest, the task gradients in Meta-ticket either stop decreasing or even increase, while ones in MAML converge to nearly zero in contrast. These results indicate that *Meta-ticket prefers rapid learning rather than feature reuse.*

In addition to the above empirical observation, we further investigate why Meta-ticket prefers rapid learning by roughly theoretical argument. For simplicity, we assume that the inner optimization consists of a single gradient step ($S = 1$). Now we consider the situation that MAML for $k$-shot learning converges, i.e. the meta-gradients for Eq. (1) with respect to $\phi_0$ in Eq. (2) approximately vanish as follows:

$$\mathbb{E}_{\mathcal{D}^{\text{tr}}, \mathcal{D}^{\text{val}} \sim \mathcal{T}^k} \Big[ \big\| \nabla_{\phi_0} \mathcal{L}(f_{\phi_0 - \alpha \nabla \mathcal{L}(f_\phi; \mathcal{D}^{\text{tr}})}; \mathcal{D}^{\text{val}}) \big\| \Big] \leq \varepsilon, \tag{5}$$

for some $\varepsilon > 0$. By Taylor expansion at $\alpha = 0$, we have $\nabla_{\phi_0} \mathcal{L}(f_{\phi_0 - \alpha \nabla \mathcal{L}(f_\phi; \mathcal{D}^{\text{tr}})}; \mathcal{D}^{\text{val}}) = \nabla_{\phi_0} \mathcal{L}(f_{\phi_0}; \mathcal{D}^{\text{val}}) + o(\alpha)$. Now we assume $|\alpha| \ll 1$ and approximate Eq. (5) by

$$\mathbb{E}_{\mathcal{D}^{\text{val}} \sim \mathcal{T}^k} \Big[ \big\| \nabla_{\phi_0} \mathcal{L}(f_{\phi_0}; \mathcal{D}^{\text{val}}) \big\| \Big] \leq \varepsilon. \tag{6}$$

By simply replacing the symbol $\mathcal{D}^{\text{val}}$ in Eq. (6) by $\mathcal{D}^{\text{tr}}$, it follows that

$$\mathbb{E}_{\mathcal{D}^{\text{tr}} \sim \mathcal{T}^k} \Big[ \big\| \nabla_{\phi_0} \mathcal{L}(f_{\phi_0}; \mathcal{D}^{\text{tr}}) \big\| \Big] \leq \varepsilon. \tag{7}$$

The left-hand side of Eq. (7) is nothing but the norm of task gradients in MAML. In other words, when the meta-gradients of MAML converges to zero during meta-training, the task gradients too.

On the other hand, if we consider the situation that Meta-ticket converges, we have

$$\mathbb{E}_{\mathcal{D}^{\mathrm{tr}}, \mathcal{D}^{\mathrm{val}} \sim \mathcal{T}^k} \Big[ \big\| \nabla_{\mathbf{m}} \mathcal{L}(f_{\mathbf{m} \odot (\phi_0 - \alpha \nabla \mathcal{L}(f_{\mathbf{m} \odot \phi}; \mathcal{D}^{\mathrm{tr}}))}; \mathcal{D}^{\mathrm{val}}) \big\| \Big] \leq \varepsilon. \tag{8}$$

By the same argument as Eq. (5–7) and by Leibniz rule, we have

$$\mathbb{E}_{\mathcal{D}^{\mathrm{tr}} \sim \mathcal{T}^k} \Big[ \big\| \nabla_{\mathbf{m}} \mathcal{L}(f_{\mathbf{m} \odot \phi_0}; \mathcal{D}^{\mathrm{tr}}) \big\| \Big] = \mathbb{E}_{\mathcal{D}^{\mathrm{tr}} \sim \mathcal{T}^k} \Big[ \big\| \phi_0 \odot \nabla_{\phi = \mathbf{m} \odot \phi_0} \mathcal{L}(f_\phi; \mathcal{D}^{\mathrm{tr}}) \big\| \Big] \leq \varepsilon. \tag{9}$$

The last inequality indicates that, if every absolute values of components in $\phi_0$ is smaller than 1 (e.g. when using Kaiming initialization), the norm of the task gradients $\nabla_\phi \mathcal{L}(f_{\mathbf{m} \odot \phi}; \mathcal{D}^{\mathrm{tr}})$ in Meta-ticket can be larger than $\varepsilon$. This is contrary to the MAML case (Eq. (7)) where the task gradients should be smaller than $\varepsilon$. In conclusion, the empirical observation that Meta-ticket prefers rapid-learning (Figure 2) can be explained by the Taylor expansion analysis under the assumption of the convergence of meta-gradients.

### 3.3  Some notes on Meta-ticket

Finally, we note some implementation details of Meta-ticket used in our experiments (Section 4).

**Parameters that should not be pruned**  In typical NN architectures, some kind of parameters should not be pruned by Meta-ticket. One type is non-matrix parameters (e.g. bias terms in linear/convolutional layers, element-wise scaling parameters in Batch/Layer normalization layers and etc). They are often unsuitable for being pruned since pruning them only hurts the adaptation capability of the NN. Another type is the parameters of the output layer for classification tasks. Pruning them means that choosing which neurons in the previous layer can contribute to some $i$-th output of the output layer. However, unlike standard supervised learning setting, $i$-th output can represent an arbitrary class in meta-learning setting. Therefore, pruning the output layer never reflect useful information for the classification tasks and only hurts adaptation capability of the NN. For the parameter $\phi_i$ of these types, we initialize and fix the corresponding mask $\mathbf{m}_i = 1$ (i.e. such $\phi_i$ is totally not pruned) throughout the whole Meta-ticket algorithm.

**Boosting approximation capacity for small NNs**  As known in the studies of strong lottery ticket hypothesis, pruning randomly initialized NNs results in less approximation capacity than standard training by directly optimizing parameters. Similarly, when the size of a given NN is small, Meta-ticket tends to achieve degraded results as we will see in Section 4. For such cases, however, we can adopt the iterative randomization technique (IteRand [9]) proposed in the lottery ticket literature, which enables us to prune randomly initialized NNs with the same level of approximation capacity as standard training. We will refer the algorithm combining Meta-ticket with the iterative randomization as **Meta-ticket+IteRand** in our experiments (Section 4.1.1).

## 4  Experiments

In this section, we conduct experiments to evaluate meta-generalization performance of Meta-ticket on multiple benchmarks of few-shot image classification. As baseline methods, we choose MAML [13] and its variants (ANIL [50] and BOIL [44]) since Meta-ticket is formulated in a parallel way to MAML and can be considered as another approach of gradient-based meta-learning. The details on every datasets, NN architectures and hyperparameters in our experiments are given in Appendix A.

### 4.1  Experiments with MLPs

In this subsection, we investigate the properties of Meta-ticket using 5-layered multilayer perceptrons (MLPs) with the ReLU activation function and batch normalization layers [24]. Since MLPs are fully-connected neural networks (NNs) designed for general tasks, we can see how Meta-ticket performs on NNs involving little inductive biases of task information. In other

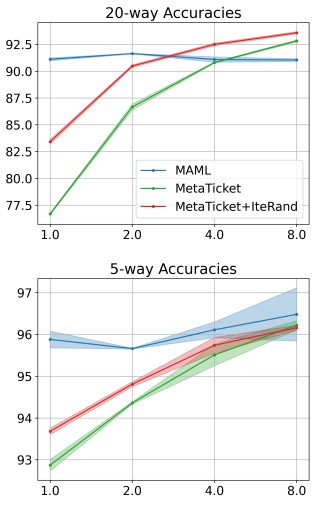

Figure 3: 5-shot accuracies on Omniglot with MLPs of various sizes. The x-axis is width factor.

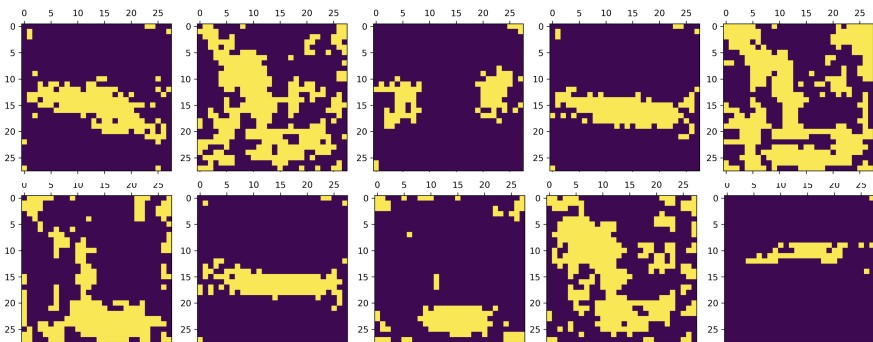

Figure 4: We visualize $28 \times 28$ binary masks $\mathbf{m}$ for the first fully-connected layer in 5-MLP with the same constant parameters. The light (resp. dark) color represents $\mathbf{m}_{ij} = 1$ (resp. $\mathbf{m}_{ij} = 0$).

Table 1: Cross-domain evaluation for 5-shot 5-way classification with 5-MLP.

| Meta-train | CIFAR-FS | | | VGG-Flower | | |
|---|---|---|---|---|---|---|
| Meta-test | CIFAR-FS | VGG-Flower | Aircraft | CIFAR-FS | VGG-Flower | Aircraft |
| MAML [13] | $53.48 \pm 0.36\%$ | $58.60 \pm 0.59\%$ | $28.95 \pm 0.76\%$ | $36.67 \pm 0.86\%$ | $59.16 \pm 0.69\%$ | $28.50 \pm 0.75\%$ |
| ANIL [50] | $50.37 \pm 0.68\%$ | $53.32 \pm 0.62\%$ | $26.86 \pm 0.77\%$ | $34.94 \pm 1.14\%$ | $58.87 \pm 1.30\%$ | $27.07 \pm 0.73\%$ |
| BOIL [44] | $55.47 \pm 1.13\%$ | $62.46 \pm 0.15\%$ | $\mathbf{33.36} \pm 0.84\%$ | $36.49 \pm 0.98\%$ | $58.15 \pm 0.86\%$ | $28.74 \pm 0.97\%$ |
| Meta-ticket | $\mathbf{55.51} \pm 0.47\%$ | $62.34 \pm 0.17\%$ | $31.85 \pm 0.31\%$ | $\mathbf{40.82} \pm 0.43\%$ | $\mathbf{64.09} \pm 0.28\%$ | $\mathbf{30.60} \pm 0.63\%$ |
| + BOIL | $\mathbf{56.06} \pm 0.57\%$ | $\mathbf{63.31} \pm 0.07\%$ | $33.02 \pm 0.32\%$ | $\mathbf{42.56} \pm 0.41\%$ | $\mathbf{65.69} \pm 0.20\%$ | $\mathbf{33.86} \pm 0.36\%$ |

words, Meta-ticket is expected to learn inductive biases from a given meta-training dataset and encode them as sparse structures in fully-connected layers of the MLPs.

We use the following small-scale datasets for meta-learning with MLPs: Omniglot [29, 59], CIFAR-FS [8], VGG-Flower [43, 30] and Aircraft [37, 30]. Omniglot is a relatively easy dataset of image classification tasks, which consists of monochrome $28 \times 28$ images of handwritten characters from different alphabets in the world. CIFAR-FS, VGG-Flower and Aircraft are also datasets of classification tasks, with more natural and colorful $32 \times 32$ images.

### 4.1.1 Experiments with various network sizes

As known in the literature [51, 49] of strong lottery tickets, in general, the pruning-only optimization like Meta-ticket tends to suffer from the lacked model capacity especially when using small NNs. To check how Meta-ticket suffers from the lacked capacity, we evaluate Meta-ticket with MLPs of various network widths, comparing to MAML as a reference. In Figure 3, we plot 5-shot meta-test accuracies on Omniglot. As we guessed in the above, Meta-ticket suffers from the degraded performance with small network widths. On the other hand, we can see that Meta-ticket even achieves superior accuracies when using larger networks on more difficult tasks of 20-way classification. Also, as we noted in Section 3.3, the results with small NNs is largely improved by applying the iterative randomization [9] (Meta-ticket+IteRand) from the lottery ticket literature.

### 4.1.2 The effect of sparse structures themselves

While Meta-ticket never directly meta-optimize the randomly initialized parameters in NNs, it can be considered that the parameter selection by Meta-ticket may produce a similar effect to the parameter optimization like MAML. Now the following question arises: *Can sparse NN structures solely be effective for few-shot learning without any information on initial parameters?* To answer this question, we consider an extremely restricted setting that all parameters are initialized with the same constant value, which prevents parameters from carrying any information about tasks. We conducted experiments in this setting on Omniglot 5-shot 5-way classification with 3 adaptation steps. The results are as follows: The subnetworks obtained by Meta-ticket achieve $71.59 \pm 1.51$ % in meta-test accuracy while random subnetworks achieve only about 33 %. This shows that Meta-ticket can find subnetworks being effective for few-shot learning even without leveraging any parameter signals. Also we visualize the sparse structures obtained by Meta-ticket in Figure 4. It indicates that the above success is due to finding sparse structures that can exploit the locality of the images in Omniglot.

Table 2: Cross-domain evaluation for 5-shot 5-way classification with large CNNs, which are meta-trained on miniImageNet and meta-tested on miniImageNet, CUB and Cars.

| Networks | ResNet-12 | | | WideResNet-28-10 | | |
|---|---|---|---|---|---|---|
| Meta-train | miniImageNet | | | miniImageNet | | |
| Meta-test | miniImageNet | CUB | Cars | miniImageNet | CUB | Cars |
| MAML | $67.47 \pm 1.31\%$ | $54.44 \pm 0.23\%$ | $43.68 \pm 1.44\%$ | $62.83 \pm 2.47\%$ | $46.46 \pm 3.54\%$ | $37.09 \pm 2.49\%$ |
| ANIL | $66.88 \pm 1.59\%$ | $53.90 \pm 1.17\%$ | $40.87 \pm 3.95\%$ | $63.01 \pm 0.82\%$ | $46.65 \pm 0.20\%$ | $37.51 \pm 1.95\%$ |
| BOIL | $69.67 \pm 0.66\%$ | $58.79 \pm 1.48\%$ | $47.11 \pm 1.10\%$ | $65.82 \pm 5.26\%$ | $55.92 \pm 0.78\%$ | $42.08 \pm 0.50\%$ |
| Meta-ticket | $\mathbf{71.31} \pm 0.29\%$ | $57.97 \pm 0.53\%$ | $45.90 \pm 0.50\%$ | $\mathbf{74.63} \pm 0.22\%$ | $\mathbf{63.26} \pm 0.86\%$ | $\mathbf{52.54} \pm 0.70\%$ |
| + BOIL | $\mathbf{74.23} \pm 0.30\%$ | $\mathbf{64.06} \pm 1.05\%$ | $\mathbf{55.20} \pm 0.64\%$ | $73.55 \pm 0.17\%$ | $62.19 \pm 0.69\%$ | $51.41 \pm 0.83\%$ |

### 4.1.3 Evaluation on different domain datasets

Finally, we evaluate the cross-domain adaptation ability of Meta-ticket with a 5-layered MLP (denoted by 5-MLP). As discussed in Section 2.3 based on the previous works [50, 44], MAML tends to suffer when adapting to tasks in unknown domains due to its feature reuse property. In contrast, since Meta-ticket prefers rapid learning rather than feature reuse (see Section 3.2), it is expected that the subnetworks obtained by Meta-ticket can generalize to tasks in unknown domains.

We conduct the cross-domain evaluation on CIFAR-FS, VGG-Flower and Aircraft. CIFAR-FS consists of classification tasks with relatively general classes from CIFAR-100 [28], and VGG-Flower and Aircraft have specialized classes. Table 1 shows the results. Meta-ticket outperforms MAML by a larger margin especially for tasks in unknown domains (i.e. different from the meta-training dataset), which indicates that Meta-ticket successfully discovered domain-generalizable sparse structures. Also, we compare with ANIL [50] and BOIL [44] as baselines, which are variants of MAML with modified inner learning rates (see Section 2.3 for details). In particular, BOIL satisfies the rapid learning property by definition and thus better meta-generalization than MAML and ANIL. In parallel to MAML, we can consider the variant of Meta-ticket with $\alpha_L = 0$ (i.e. the same modification as BOIL) which is denoted by **Meta-ticket+BOIL**. Since Meta-ticket+BOIL achieves almost the same or better accuracies than BOIL, we find out that the Meta-ticket-based method also benefits from the modification and still achieves superior meta-generalization than the MAML-based methods.

### 4.2 Experiments with CNNs on mini-Imagenet

Meta-ticket also can be applied to convolutional neural networks (CNNs). Although CNNs already have sparse NN structures as convolutional kernels, they can still benefit from the rapid learnable nature of Meta-ticket. To evaluate the practical benefits as well as meta-generalization ability, we conducted cross-domain evaluation on near real-world datasets. We employed miniImagenet dataset [59, 52], which consists of image classification tasks with $84 \times 84$ images and general classes, for meta-training/testing and specialized datasets such as classification of birds (CUB dataset [63, 21]) and cars (Cars dataset [27, 57]) for meta-testing. For CNN architectures, we chose near-practical deep ones: ResNet-12 [20] and WideResNet-28-10 [66].

The results are shown in Table 2. Meta-ticket and its BOIL-variant consistently outperform MAML-based methods, by a large margin especially when evaluated on specialized domains, probably because Meta-ticket can adjust its feature extractor to recognize fine-grained differences by the rapid learning. Moreover, while MAML struggles to exploit the potential of a larger network such as WideResNet-28-10, Meta-ticket can benefit from such scaling up of network size.

## 5 Related Work

**Gradient-based meta-learning** Meta-learning [6, 54, 55] has been studied for decades to equip computers with learning capabilities. There are several approaches depending on how the inner optimization is modeled. Black-box meta-learning approaches model the inner optimization by recurrent [22, 3, 52] or feedforward [41, 19] neural networks. Gradient-based meta-learning [14] models the inner optimization by one or multiple steps of gradient descent, which leads to sample-efficient meta-learning due to the strong inductive bias on the inner optimization. MAML [13] is a representative method of the gradient-based approach, which meta-learns initial parameters for the

inner optimization. There are also extensions of MAML, including meta-learnable learning rates [34] and transformation matrices [47, 15, 5, 60] of the inner optimization. Some works [50, 44] discussed on feature reuse of MAML, and pointed out that it may hurt the meta-generalization capacity of MAML. Since our proposed method (Meta-ticket) is another gradient-based meta-learning method formulated in parallel to MAML, most of extensions of MAML can also be applied to Meta-ticket. Finally, we note on related works involving both meta-learning and pruning. Tian et al. [56] and Gao et al. [18] investigated magnitude-based pruning methods for meta-learning of MAML, while our work focused on meta-learning by pruning only. Alizadeh et al. [2] exploits meta-gradient to prune randomly initialized neural networks. Although the methodology is very similar to ours, their target is pruning at initialization [33] for standard learning, not few-shot learning.

**Learning neural network structures from data**   Some prior works [17, 42, 68] attempt to automatically learn inductive biases on neural network structures conventionally designed by human. Neyshabur [42] pointed out that the superior generalization capacity of CNNs is mainly due to sparse structures, which enables to leverage locality of image data. Also they proposed a pruning scheme to automatically find such sparse structures, but it does not involve meta-learning and not designed for few-shot learning. Another line of research is neural architecture search (NAS [70]). In particular, meta-learning-based NAS [36, 31, 65] can efficiently search neural architectures. Also, NAS for few-shot learning [25, 35, 12, 62, 61] is closely related to our work. Since NAS aims to automatically design how to connect pre-defined blocks and not to find sparse structures in each block, it is orthogonal to our approach of finding sparse NN structures.

**Lottery ticket hypothesis**   Lottery ticket hypothesis (LTH) is originally proposed in Franckle and Carbin [16], which claims the existence of sparse subnetworks in randomly initialized NNs such that the former can achieve as good accuracy as the latter after training. Inspired by the analysis in Zhou et al. [69], Ramanujan et al. [51] proposed to train NNs by pruning randomly initialized NNs (weight-pruning optimization) without any optimization of the initial parameters (weight optimization). Their hypothesis are formulated as strong LTH: There exist sparse subnetworks in randomly initialized NNs that already achieves good accuracy *without training*. The follow-up works [38, 45, 49] theoretically proved it under assumption on the network size. From the viewpoint of LTH, Meta-ticket can be considered as a meta-learning method with weight-pruning optimization, and MAML as one with weight optimization. Surprisingly, even though weight-pruning optimization tends to achieve worse accuracies than weight optimization in general, Meta-ticket outperforms MAML by a large margin in cross-domain evaluation.

**Applications of sparse NN structures to continual learning**   Continual learning [55, 46] aims to learn new tasks sequentially without forgetting the capacity learned from past tasks. To handle such multiple tasks, some works [40, 39, 23, 64] leverage sparse NN structures in continual learning. In particular, the approach by Wortsman et al. [64] shares a similar spirit to ours in that they leverage subnetworks of a randomly initialized neural network. Unlike our approach, which finds an optimal subnetwork common to multiple tasks, they leverage task-specific subnetworks to separate learned results from multiple tasks.

## 6   Limitations

As suggested from the theory of strong lottery tickets [38, 45, 49], Meta-ticket also tends to degrade with small NNs compared to MAML (see Section 4.1.1). However, the degradation can be reduced largely by iterative randomization technique [9] as explained in Section 3.3. Moreover, Meta-ticket actually outperforms MAML when using larger NNs. We regard Meta-ticket as an alternative gradient-based meta-learning of MAML, and thus one can switch either of them to use depending on the NN scale. Also, since our theoretical analysis on task gradients (Section 3.2) heavily depends on Taylor series approximation, it holds only when the inner learning rates are sufficiently small. For general cases, it might require the arguments involving the theory on gradient-based meta-learning with SGD. Hence, proving the feature reuse in MAML still remains an open problem.

# 7 Conclusion

In this work, we investigated sparse NN structures that are effective for few-shot learning. We proposed a novel gradient-based meta-learning method called Meta-ticket, which discovers optimal subnetworks for few-shot learning within randomly initialized neural networks. While Meta-ticket is formulated in parallel to MAML, we found that Meta-ticket prefers rapid learning rather than feature reuse in contrast to MAML. As a result, since the resulting feature extractor obtained by Meta-ticket can recognize specialized features for each task, Meta-ticket can achieve superior meta-generalization especially on unknown domains. Moreover, we found that Meta-ticket can benefit from scaling up of the size of neural networks while MAML struggles to exploit the potential of large NNs. We hope that this work opens up a new research direction in gradient-based meta-learning, focusing on sparse NN structures rather than initial parameters.

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
