# OpenReview forum: "Meta-ticket: Finding optimal subnetworks for few-shot learning within randomly initialized neural networks"
_NeurIPS.cc/2022/Conference — NeurIPS 2022 Accept_

### Official Review · Reviewer_KHZB · 2022-07-03

**Rating:** 5
**Confidence:** 3
**Soundness:** 3 good
**Presentation:** 3 good
**Contribution:** 3 good

**Summary:**

The authors propose Meta-ticket as a meta-learning approach that finds optimal sparse subnetworks for few-shot learning within the randomly initialized neural network. The paper can be seen as a MAML modification.

**Questions:**

1. How does the method work on the pretrain model?
2. Is it possible to train the weight of the network simultaneously with mask m?
3. It is a good way to compare original MAML and Meta-ticket with some architectures. Does it seem that MAML requires fewer parameters to solve a few-shot task than Meta-ticket?

**Ethics Review Area:**

["I don’t know"]

**Limitations:**

The limitation section is well written.

**Strengths And Weaknesses:**

Strengths
1. The paper is well written and easy to follow.
2. The idea looks fresh and interesting.
3. The analysis of rapid learning and feature reuse is well done.
Weaknesses
1. It is unclear why the authors refer only to MAML (and its modification). If the paper introduces a new method for a few shots, the method should be compared with the state-of-the-art model using the gaussian process, transformers, and hyper networks.
2. On the other hand, the paper can be seen as another MAML-based method analysis. But the contribution in the paper should be changed.
3. It is not clear why authors use randomly initialized neural networks. How does the method work on the pretrain model?
4. The fig. 1 should be better described.
5. Section 3.1 is not clear. Functions CalculateMask() and UpdateScore() are only mentioned and do not described in detail.

---

> ### Author Response · Authors · 2022-08-01
> **Thank you for your review (1/2)**
>
> Thanks for your valuable review and suggestions.
>
> ### Methods for comparison
>
> >(Strengths and Weaknesses 4.) It is unclear why the authors refer only to MAML (and its modification). If the paper introduces a new method for a few shots, the method should be compared with the state-of-the-art model using the gaussian process, transformers, and hyper networks.
> >
> >(Strengths and Weaknesses 5.) On the other hand, the paper can be seen as another MAML-based method analysis. But the contribution in the paper should be changed.
>
> First of all, to explain why we mainly use MAML-based methods for comparison, we would like to recall the purpose of our paper along with Introduction (Section 1).
> **The purpose of our paper is to answer the following research questions**: `(1) Can we automatically find sparse NN structures that are effective for few-shot learning by meta-learning? (2) What are the strengths and weaknesses of the meta-learned NN structures compared to the meta-learned initial parameters of MAML? (lines 48-50)`
> To answer the question (1), we introduced Meta-ticket as a novel gradient-based meta-learning approach, where we directly formulated finding `sparse NN structures that are effective for few-shot learning by meta-learning` in Section 3.1.
> **As a result, the formulation of Meta-ticket becomes very parallel to the formulation of MAML**.
> From this perspective, we can see that the only difference between Meta-ticket and MAML is their meta-parameters; the former meta-optimizes the mask parameter $m$ and the latter meta-optimizes the initial parameter of given NNs.
> Therefore, **in order to evaluate the meta-generalization ability of sparse NN structures in a fair way**, we decided to use MAML and its variants for fair comparison, i.e. the same network architecture, the same number of meta-parameters and the same hyperparameters except for meta-optimization.
>
> However, even though our purpose or experimental design is not for achieving state-of-the-art performance as explained above, we agree that the comparison with other state-of-the-art methods is also helpful to understand how effective our approach is.
> Thus we add Table 8 in the revised version of Appendix (Section B.4) to compare Meta-ticket with such state-of-the-art methods.
> Here we summarize Table 8 as follows:
>
> |    Methods  | miniImageNet -> miniImageNet | miniImageNet -> CUB | miniImageNet -> Cars |
> | :---    | :---: | :---: | :---: |
> |  Meta-ticket (Ours) |  $71.31 \pm 0.29\\%$ | $57.97 \pm 0.53 \\%$  | $45.90 \pm 0.50 \\%$ |
> |  Meta-ticket + BOIL (Ours) | $74.23 \pm 0.30 \\%$ | $64.06 \pm 1.05 \\%$ | $\mathbf{55.20 \pm 0.64} \\%$ |
> |  MetaOptNet-SVM-trainval [12] | $80.00 \pm 0.45 \\%$ | $54.67 \pm 0.56 \\%$ | $45.90 \pm 0.49 \\%$ |
> |  GNN + Feature-wise Transformation [20] | $\mathbf{81.98 \pm 0.55} \\%$ | $\mathbf{66.98 \pm 0.68} \\%$ | $44.90 \pm 0.64 \\%$ |
>
> (The results of MetaOptNet and Feature-wise Transformation are cited from Tseng et al. [20].)
>
>
> Interestingly, **even though we did not design or tune Meta-ticket** for cross-domain adaptation, it achieves **similar performance** (on miniImageNet -> CUB) or even **largely outperforms** these methods (on miniImageNet -> Cars) on the cross-domain evaluation.
> Since these results indicate the strength of sparse NN structures for few-shot learning, we are interested in future work to achieve SOTA performance based on our approach.
>
> ### Applications to the pretrained model
>
> > (Strengths and Weaknesses 6.) It is not clear why authors use randomly initialized neural networks. How does the method work on the pretrain model?
> >
> > (Questions 1.) How does the method work on the pretrain model?
>
> The reason of why we use randomly initialized NNs is mainly to evaluate sparse NN structures themselves for few-shot learning, without additional weight-optimization like MAML.
> Even so, we agree that how Meta-ticket behaves on the pretrained model is an interesting question.
> To check this, we conducted additional experiments of applying Meta-ticket to the model pretrained by MAML.
> The results are as follows:
>
> | Methods | miniImageNet -> miniImageNet | miniImageNet -> CUB | miniImagenet -> Cars |
> | :--- | :---: | :---: | :---: |
> | MAML pretraining -> Meta-ticket | $70.78 \pm 0.35 \\%$ | $55.28 \pm 0.22 \\%$ | $44.88 \pm 0.23 \\%$ |
> | MAML (from our paper) | $67.47 \pm 1.31 \\%$ | $54.44 \pm 0.23 \\%$ | $43.68 \pm 1.44 \\%$ |
> | Meta-ticket (from our paper) | $71.31 \pm 0.29 \\%$ | $57.97 \pm 0.53 \\%$ | $45.90 \pm 0.50 \\%$ |
>
> Although it outperforms MAML, the results indicate that pruning pretrained networks is actually worse than pruning randomly initialized networks.
> This may be because (i) **the parameters pretrained by MAML are mainly optimized for feature reuse and thus they go wrong with the rapid learning nature of Meta-ticket** and (ii) **the pretrained parameters are less uniformly distributed than randomly initialized parameters**, which leads to sub-optimal solutions in the later Meta-ticket phase.

---

> > ### Author Response · Authors · 2022-08-01
> > **Thank you for your review (2/2)**
> >
> >
> > ### Joint training with MAML
> >
> > > (Questions 2.) Is it possible to train the weight of the network simultaneously with mask m?
> >
> > This is also an interesting question.
> > We conducted additional experiments to answer this question, by meta-optimizing the mask parameter $m$ (by Meta-ticket) and the initial parameter $\phi_0$  (by MAML) simultaneously.
> > The results are the following:
> >
> > | Methods | miniImageNet -> miniImageNet | miniImageNet -> CUB | miniImagenet -> Cars |
> > | :--- | :---: | :---: | :---: |
> > | MAML + Meta-ticket | $67.57 \pm 1.61 \\%$ | $53.85 \pm 0.78 \\%$ | $41.61 \pm 1.59 \\%$ |
> > | MAML (from our paper) | $67.47 \pm 1.31 \\%$ | $54.44 \pm 0.23 \\%$ | $43.68 \pm 1.44 \\%$ |
> > | Meta-ticket (from our paper) | $71.31 \pm 0.29 \\%$ | $57.97 \pm 0.53 \\%$ | $45.90 \pm 0.50 \\%$ |
> >
> > The results of the joint training (MAML + Meta-ticket) is very similar to MAML and thus degrades than Meta-ticket alone, even though the resulting network is actually sparsified by Meta-ticket.
> > Moreover, it is remarkable that **MAML + Meta-ticket behaves similarly to MAML even in cross-domain adaptation**.
> > This is because the feature reuse nature of MAML (as discussed in Section 3.2) dominates the meta-optimization if we jointly use MAML and Meta-ticket.
> >
> > ### Limitations of Meta-ticket
> >
> > > (Questions 3.) Does it seem that MAML requires fewer parameters to solve a few-shot task than Meta-ticket?
> >
> > Partially yes, particularly with small neural networks.
> > In Limitations section (Section 6), we already discussed this as follows: `Meta-ticket tends to degrade with small NNs compared to MAML (see Section 4.1.1). However, the degradation can be reduced largely by iterative randomization technique [6] as explained in Section 3.3.`
> >
> > On the other hand, this limitation turns out to be an **advantage** of our approach when we want to leverage the capacity of larger networks.
> > As shown in Section 4.1.1 & 4.2, while MAML-based methods struggle with leveraging the scaling merits of the network size, Meta-ticket successfully leverage them.
> >
> > [6] Chijiwa et al., "Pruning randomly initialized neural networks with iterative randomization." (NeurIPS 2021)
> >
> > ### Explanations
> >
> > > (Strengths and Weaknesses 7.) The fig. 1 should be better described.
> >
> > Thanks for pointing out that. In the final version of the paper, we will improve the caption of Figure 1 to be more precise as follows:
> >
> > `(Before) Meta-ticket is a novel gradient-based meta-learning approach. For a randomly initialized NN, instead of optimizing the initial parameters, Meta-ticket finds an optimal subnetwork that can adapt to given few-shot tasks without overfitting.`
> >
> > `(After) For a randomly initialized neural network $f_\phi(x)$, instead of optimizing the initial parameters $\phi$, Meta-ticket finds an optimal subnetwork $f_{\vect{m} \odot\phi}(x)$ by meta-optimizing binary parameters $\vect{m}$ so that the subnetwork can successfully adapt to tasks with a few samples.`
> >
> >
> > > (Strengths and Weaknesses 8.) Section 3.1 is not clear. Functions CalculateMask() and UpdateScore() are only mentioned and do not described in detail.
> >
> > Actually, in Section 3.1, we gave the precise descriptions of them with mathematical expressions.
> > Please see lines 149-155 and Eq. (4) for CalculateMask(s) and lines 156-159 for UpdateScore().

---

> > ### Comment · Reviewer_KHZB · 2022-08-05
> > **(Strengths and Weaknesses 4.)**
> >
> >  I understand that the paper's primary goal is answering the above questions, but in my opinion, the model should be compared with the state-of-the-art solution (in the main paper). Mainly not only MAML based. Furthermore, many MAML modifications exist, like MAML++, Reptile, iMAML-HF, and Meta-SGD to compare.

---

> > > ### Author Response · Authors · 2022-08-08
> > > **Thank you for your reply to our rebuttal**
> > >
> > > Thank you for the additional explanation of your concern.
> > >
> > > > but in my opinion, the model should be compared with the state-of-the-art solution (in the main paper). Mainly not only MAML based.
> > >
> > > The methods added in the rebuttal, MetaOptNet and Feature-wise Transformation, are actually the state-of-the-art methods for cross-domain classification and are not based on MAML.
> > > Also, even though the comparison with such SOTA methods helps us to see the practicality of our method, it is difficult to disentangle what causes the difference in accuracy between these SOTA methods and ours.
> > > Unlike the controlled experiments with MAML (Section 4), such a naive comparison with SOTA methods provides less scientific insight into the nature of the sparse NN structures (which is the main question in our paper).
> > > Therefore, we still think it would be better to include the extended comparison results into the appendix, so that readers are not confused unnecessarily.
> > >
> > > > Furthermore, many MAML modifications exist, like MAML++, Reptile, iMAML-HF, and Meta-SGD to compare.
> > >
> > > First of all, we would like to emphasize that Meta-ticket should be seen as an alternative of MAML, rather than a MAML modification.
> > > It means that **many MAML modifications (including BOIL [32] as shown in Section 4) can also be applied to Meta-ticket in a parallel way to MAML**.
> > > For example, we can apply Meta-SGD [24] to improve the accuracy of Meta-ticket as follows:
> > >
> > > | Methods (5-shot 5-way) | miniImageNet -> miniImageNet | miniImageNet -> CUB | miniImageNet -> Cars |
> > > | :--- | :---: | :---: | :---: |
> > > | MAML | $67.47 \pm 1.31 \\%$ | $54.44 \pm 0.23 \\%$ | $43.68 \pm 1.44\\%$ |
> > > | MAML + **Meta-SGD** | $69.63 \pm 1.50 \\%$ | $58.53 \pm 1.69  \\%$ | $49.14 \pm 1.29  \\%$ |
> > > |  Meta-ticket |  $71.31 \pm 0.29\\%$ | $57.97 \pm 0.53 \\%$  | $45.90 \pm 0.50 \\%$ |
> > > | Meta-ticket + **Meta-SGD** | $\mathbf{72.11 \pm 0.56}  \\%$ | $\mathbf{61.13 \pm 0.46}  \\%$ | $\mathbf{49.43 \pm 0.52} \\%$ |
> > >
> > > This is just an example and there are a lot of other modifications, but most of them should be also applicable to Meta-ticket since Meta-ticket has been formulated parallelly to MAML.
> > >
> > > ### References
> > >
> > > [24] Li et al. "Meta-SGD: Learning to learn quickly for few-shot learning." (arXiv:1707.09835)
> > >
> > > [32] Oh et al. "BOIL: Towards representation change for few-shot learning." (ICLR 2021)

---

### Official Review · Reviewer_xhDJ · 2022-07-11

**Rating:** 6
**Confidence:** 4
**Soundness:** 3 good
**Presentation:** 3 good
**Contribution:** 3 good

**Summary:**

This works presents a few-shot learning framework named Meta-Ticket based on gradient-based meta learning. The meta learning is applied to select optimal sub-network from a randomly initialised network, so it’s easier to transfer to a novel domain, as similar to MAML. Meta-ticket has shown to achieve comparable, most of the time better performance than MAML, particularly in larger NN setting, and with more difficult datasets. The paper additionally provides interesting discussions on learning behaviour of few-shot adaptation in comparison with MAML, both in empirical and theoretical perspective.

**Questions:**

See the weaknesses.

**Strengths And Weaknesses:**

The paper is very easy to follow and supported with a clear and interesting motivation. In general, the authors present a very novel way of doing few-shot learning, combing some of different research areas of machine learning of Lottery ticket hypothesis, randomly initialised network and meta learning.  The discussion on rapid learning v.s. feature reuse is one hightlight of this work, to better understand the learning strategies of different few-shot learning methods. The authors also support the observation with theoretical argument.

Some additional comments which may further improve this work:

-- Search the optimal sub-network is a well-studied technique in transfer/multi-task learning strategy. I would suggest adding relevant related works and slightly discuss the difference of the proposed mask searching strategy: [1,2,3].

[1] Mallya et al. Piggyback: Adapting a Single Network to Multiple Tasks by Learning to Mask Weights, 2018.

[2] Hung et al. Compacting, Picking and Growing for Unforgetting Continual Learning 2019.

[3] Wortsman et al. Supermasks in Superposition 2020.

-- After searching the optimal subnetwork, I assume the rest of the network parameters are remained unused? If so, the baseline methods were trained with a larger parameter size? This actually leads to unfair comparison, as in the few-shot learning literature, it may leads to degraded performance due to overfitting.

-- Is there any hyper-parameter controlling the size of the subnetwork? It would be interesting to see how different size of subnetwork can relate to the final few-shot learning performance.

---

> ### Author Response · Authors · 2022-08-01
> **Thank you for your review**
>
> Thanks for your valuable suggestions and questions.
>
> ### Additional related work
>
> > -- Search the optimal sub-network is a well-studied technique in transfer/multi-task learning strategy. I would suggest adding relevant related works and slightly discuss the difference of the proposed mask searching strategy: [1,2,3].
>
> Thank you for sharing additional related work.
> We agree that the discussion on the suggested litrature is also helpful to understand our approach.
> In a nutshell, the main differences between these literature and our approach are (i) they are intended for improving continual/incremental learning rather than few-shot learning and (ii) their methods do not involve meta-optimization as Meta-ticket does.
> We will discuss on these differences in Related Work section of the final version of the paper.
>
> ### Parameter size in inner loops
>
> > -- After searching the optimal subnetwork, I assume the rest of the network parameters are remained unused? If so, the baseline methods were trained with a larger parameter size? This actually leads to unfair comparison, as in the few-shot learning literature, it may leads to degraded performance due to overfitting.
>
> Yes, the pruned parameters in Meta-ticket are unused in the inner optimization during meta-training/meta-testing.
> Thus, the number of parameters for adaptation (here we call them *inner parameters* for simplicity) in Meta-ticket is less than in MAML.
>
> However, we consider that such inconsistency of the number of inner parameters is not unfair setting.
> This is because, while there are high correlation between the number of meta-parameters (i.e., parameters that are optimized in outer optimization) and the meta-test accuracy, but there are less correlation between the number of inner parameters and the meta-test accuracy.
> For example, ANIL [36] has negligibly fewer inner parameters than MAML, since only the parameters in the classification layer can adapt in ANIL.
> However, as we can see in Table 1 and 2 (Section 4), ANIL behaves very similarly to or even worse than MAML, whenever the number of meta-parameters is same.
> This indicates that the less number of inner parameters does not simply imply the improved performance.
>
> Even so, we agree that the following question is interesting: What if the inner parameters in MAML are also pruned?
> Related to the [question (2) from Reviewer KHZB](https://openreview.net/forum?id=Cr4_3ptitj&noteId=D_GSZvuUrV), we conducted additional experiments where both of the initial parameter (MAML) and the mask parameter (Meta-ticket) are jointly meta-trained.
> The results are as follows:
>
> | Methods | miniImageNet -> miniImageNet | miniImageNet -> CUB | miniImagenet -> Cars |
> | :--- | :---: | :---: | :---: |
> | MAML + Meta-ticket | $67.57 \pm 1.61 \\%$ | $53.85 \pm 0.78 \\%$ | $41.61 \pm 1.59 \\%$ |
> | MAML (from our paper) | $67.47 \pm 1.31 \\%$ | $54.44 \pm 0.23 \\%$ | $43.68 \pm 1.44 \\%$ |
> | Meta-ticket (from our paper) | $71.31 \pm 0.29 \\%$ | $57.97 \pm 0.53 \\%$ | $45.90 \pm 0.50 \\%$ |
>
> While the resulting network of the joint meta-training (MAML + Meta-ticket) is almost as sparse as Meta-ticket, the meta-test accuracy is similar to or even worse than MAML.
> This is because the feature reuse nature of MAML (as discussed in Section 3.2) dominates the meta-optimization if MAML and Meta-ticket are jointly used.
> In conclusion, these results indicate that **the success of Meta-ticket may be due to its meta-optimization, rather than the number of inner parameters**.
>
> [36] Raghu et al. "Rapid learning or feature reuse? Towards understanding the effectiveness of MAML." (ICLR 2020)
>
> ### How to control the final sparsity
>
> > -- Is there any hyper-parameter controlling the size of the subnetwork? It would be interesting to see how different size of subnetwork can relate to the final few-shot learning performance.
>
> Yes, we can control the final sparsity by appropriately setting the initial sparsity hyperparameter $p_{init} \in [0,1]$ (See Section 3.1 for the definition). As we can see in Appendix B.2, there are (1) **a correlation between the initial sparsity $p_{init}$ and the final sparsity of the subnetwork** (Figure 1 in Appendix B.2) and (2) **a trade-off between the final sparsity and the final accuracy** (Table 6 in Appendix).
>
> Therefore, we can control the trade-off between the final sparsity and the final accuracy by varying the initial sparsity $p_{init}$. Finally, note that we employed $p_{init} = 0.0$ (which leads to obtain a less sparse subnetwork) for all experiments in the paper (except for some ablation studies) in order to maximize the final accuracy.

---

> > ### Comment · Reviewer_xhDJ · 2022-08-05
> > **Response to rebuttal**
> >
> > Thanks for the authors for the rebuttal. I have no further questions and would like to keep the original weak accept as my final rating.

---

### Official Review · Reviewer_hLtC · 2022-07-12

**Rating:** 5
**Confidence:** 5
**Soundness:** 3 good
**Presentation:** 3 good
**Contribution:** 2 fair

**Summary:**

This paper introduces a novel gradient-based meta-learning method, Meta-Ticket, as an alternative of MAML. For meta-objective, Meta-Ticket aims to optimize a sparse subnetwork hidden within a randomly initialized neural network (meta-mask, called meta-ticket), while MAML optimizes the initial parameters (meta-initialization).

The experiments show that, Meta-ticket can achieve superior performance than MAML on cross-domain few-shot settings (over different datasets), although it tends to achieve worse accuracies than MAML in general setting.

**Questions:**

If ProsPr [1] is applied to a single few-shot task, can it achieve the same results as Meta-ticket (that is trained over many few-shot tasks) on testing few-shot tasks?

**Limitations:**

See above

**Strengths And Weaknesses:**

Pros:
1. If focusing on cross-domain few-shot applications, the proposed Meta-Ticket achieves better meta-generalization ability rather than MAMLs.
2. Sufficient experiments over many datasets (domains) focusing on different cross-domain tasks and different types of NNs.

Cons:
1. Novelty is limited. The proposed Meta-Ticket algorithm is almost the same as ProsPr [1], which also optimizes a subnetwork (learns a mask $\mathbf{m}$) AND uses meta-gradients to take into account that we are going to train the weights after we apply mask to get subnetwork.

However, there are two differences between this paper and [1]:
    (1) The x <- UpdateScores(s, L) in the paper replaces the "saliency scores computation" in [1] with the technique in [37].
    (2) The training steps after pruning, in this paper, is obtained from the few-shot validation data from multiple tasks. But in [1], the training steps after pruning is standard training from large data of single task.

Despite the two differences, I still think the novelty of the proposed algorithm is not enough.

2. As pointed out in the paper (Section 6 and 4.1.1), the Meta-Ticket may tend to optimize larger network to achieve similar results as MAML, as it tries to find a hidden prune-mask within the fixed random parameters. This could be a limitation (i.e., memory problem, speed issue, etc) if training machines do not have enough resources.
3. In experiments, missing a lot of SOTA gradient-based meta-learning methods. Especially, there are many task-specific gradient-based methods (e.g., LEO, MMAML) that is good for cross-domain few-shot learning. Maybe the authors should consider including them.

[1] Milad Alizadeh, Shyam A. Tailor, Luisa M Zintgraf, Joost van Amersfoort, Sebastian Far- quhar, Nicholas Donald Lane, and Yarin Gal. Prospect pruning: Finding trainable weights at initialization using meta-gradients. In International Conference on Learning Representations, 2022.

[37] Vivek Ramanujan, Mitchell Wortsman, Aniruddha Kembhavi, Ali Farhadi, and Mohammad Rastegari. What’s hidden in a randomly weighted neural network? In Proceedings of the IEEE/CVF Conference on Computer Vision and Pattern Recognition, pages 11893–11902, 2020.

---

> ### Author Response · Authors · 2022-08-01
> **Thank you for your review (1/2)**
>
> Thanks for your valuable review and suggestions.
>
> ### The novelty in our paper
>
> > 1. Novelty is limited. The proposed Meta-Ticket algorithm is almost the same as ProsPr [1], which also optimizes a subnetwork (learns a mask m) AND uses meta-gradients to take into account that we are going to train the weights after we apply mask to get subnetwork. However, there are two differences between this paper and [1]: (...) Despite the two differences, I still think the novelty of the proposed algorithm is not enough.
>
> First of all, we would like to emphasize the purpose and contributions of our paper, which are described in Introduction (Section 1).
> The purpose of our paper is to answer the following research questions: `(1) Can we automatically find sparse NN structures that are effective for few-shot learning by meta-learning? (2) What are the strengths and weaknesses of the meta-learned NN structures compared to the meta-learned initial parameters of MAML? (lines 48-50 in Section 1)`
> The contributions of our paper is answering these questions by (i) introducing a novel gradient-based meta-learning formulation as Meta-ticket, (ii) evaluating the meta-generalization ability of the sparse subnetwork found by Meta-ticket in a fair experimental setting, and (iii) analyzing why Meta-ticket can achieve higher meta-generalization ability particularly in cross-domain setting.
> **All of these contributions should be considered as part of the novelty of our paper.**
>
> Nevertheless, we also consider Meta-ticket itself as a novel meta-learning algorithm for few-shot learning, even though it is similar to ProsPr [1] as already mentioned in Related Work section (Section 5).
> ProsPr is a meta-gradient-based method for the problem of pruning at initialization, which aims to find a sparse subnetwork that can be trained **without hurting its training dynamics**.
> More precisely, ProsPr identifies the unnecessary parameters that can be eliminated without loss of training accuracy, using meta-gradients of the training loss after several gradient steps.
> On the other hand, Meta-ticket is designed to find a sparse subnetwork that can learn from a novel task with limited data **without overfitting**.
>
> As a result, **Meta-ticket has important differences from ProsPr**: Meta-ticket optimizes both the mask parameter $m$ and its sparsity rate using multiple tasks so that it achieves high test accuracy after gradient steps for each task. In contrast, ProsPr **does not** optimize any parameters or sparsity rate; it only computes saliency scores using (the absolute values of) meta-gradients of training loss after gradient steps for several batches from a single task, which successfully leads to its lightweight computation for pruning at initialization.
>
> ### Limitation with small neural networks
>
> > 2. As pointed out in the paper (Section 6 and 4.1.1), the Meta-Ticket may tend to optimize larger network to achieve similar results as MAML, as it tries to find a hidden prune-mask within the fixed random parameters. This could be a limitation (i.e., memory problem, speed issue, etc) if training machines do not have enough resources.
>
> In Limitations section (Section 6), we already discussed this as follows: `Meta-ticket tends to degrade with small NNs compared to MAML (see Section 4.1.1). However, the degradation can be reduced largely by iterative randomization technique [6] as explained in Section 3.3. Moreover, Meta-ticket actually outperforms MAML when using larger NNs. We regard Meta-ticket as an alternative gradient-based meta-learning of MAML, and thus one can switch either of them to use depending on the NN scale.`
> Therefore, we can handle even such situation with restricted computational resources.
>
> Also, this limitation turns out to be an **advantage** of our approach when we want to leverage the capacity of larger networks.
> As shown in Section 4.1.1 & 4.2, while MAML-based methods struggle with leveraging the scaling merits of the network size, Meta-ticket successfully leverage them.
>
>
> ## References
>
> [1] Alizadeh et al., "Prospect pruning: Finding trainable weights at initialization using meta-gradients." (ICLR 2022)
>
> [6] Chijiwa et al., "Pruning randomly initialized neural networks with iterative randomization." (NeurIPS 2021)

---

> > ### Author Response · Authors · 2022-08-01
> > **Thank you for your review (2/2)**
> >
> > ### Comparison with SOTA methods
> >
> > > 3. In experiments, missing a lot of SOTA gradient-based meta-learning methods. Especially, there are many task-specific gradient-based methods (e.g., LEO, MMAML) that is good for cross-domain few-shot learning. Maybe the authors should consider including them.
> >
> > Even though our purpose is not to achieve state-of-the-art performance as explained above, we agree that the comparison with other state-of-the-art methods is also helpful to understand how effective our approach is.
> > We add Table 8 in the revised version of Appendix to compare Meta-ticket with state-of-the-art methods [12, 20] for cross-domain classification.
> > Here we summarize Table 8 as follows:
> >
> > |    Methods    | miniImageNet -> miniImageNet | miniImageNet -> CUB | miniImageNet -> Cars |
> > | :---          | :---: | :---: | :---: |
> > |  MAML  | $67.47 \pm 1.31 \\%$  | $54.44 \pm 0.23 \\%$  | $43.68 \pm 1.44 \\%$  |
> > |  BOIL    | $69.67 \pm 0.66 \\%$  | $58.79 \pm 1.48 \\%$  | $47.11 \pm 1.10 \\%$  |
> > |  Meta-ticket (**Ours**) |  $71.31 \pm 0.29\\%$ | $57.97 \pm 0.53 \\%$  | $45.90 \pm 0.50 \\%$ |
> > |  Meta-ticket + BOIL (**Ours**) | $74.23 \pm 0.30 \\%$ | $\mathbf{64.06 \pm 1.05} \\%$ | $\mathbf{55.20 \pm 0.64} \\%$ |
> > |  MetaOptNet-SVM-trainval [12] | $80.00 \pm 0.45 \\%$ | $54.67 \pm 0.56 \\%$ | $45.90 \pm 0.49 \\%$ |
> > |  GNN + Feature-wise Transformation [20] | $\mathbf{81.98 \pm 0.55} \\%$ | $\mathbf{66.98 \pm 0.68} \\%$ | $44.90 \pm 0.64 \\%$ |
> >
> > (The results of MetaOptNet and Feature-wise Transformation are cited from Tseng et al. [20].)
> >
> > Actually, both of these SOTA methods achieve higher meta-test accuracy on miniImageNet itself, since they can leverage their strong feature extractor on the meta-training dataset.
> > On the other hand, when evaluated on the other datasets such as CUB and Cars, Meta-ticket + BOIL achieves **similar performance** (on miniImageNet -> CUB) or even **largely outperforms** by $+10 \\%$ (on miniImageNet -> Cars) these SOTA methods.
> > As these results also indicate **the effective meta-generalization ability of sparse NN structures**, we are interested in future work to achieve SOTA performance based on our approach.
> >
> > ### Comparison with ProsPr applied to few-shot learning
> >
> > > Questions: If ProsPr [1] is applied to a single few-shot task, can it achieve the same results as Meta-ticket (that is trained over many few-shot tasks) on testing few-shot tasks?
> >
> > We conducted additional experiments to answer this question.
> > We apply ProsPr to a single or multiple few-shot tasks and evaluate the resulting pruned network (on 5-show 5-way miniImageNet).
> > The results are as follows:
> >
> > | (Initial) Sparsity             | 0.0 | 0.3 | 0.5 | 0.7 | 0.9 |
> > | :---          | :---: | :---: | :---: | :---: | :---: |
> > | ProsPr (1 task) | $26.77 \pm 3.24  \\%$ | $26.71 \pm 2.82  \\%$ | $24.51 \pm 2.41  \\%$ | $22.31 \pm 2.77   \\%$ | $22.58 \pm 2.59   \\%$ |
> > | ProsPr (64 tasks) | $26.77 \pm 3.24   \\%$ | $26.82 \pm 2.79   \\%$ | $25.43 \pm 2.46   \\%$ | $23.14 \pm 2.47   \\%$ | $22.34 \pm 2.10   \\%$ |
> > | Random masks | $26.82 \pm 1.79 \\%$ | $28.01 \pm 2.41  \\%$ |  $27.99 \pm 2.93 \\%$  |$28.94 \pm 2.95 \\%$ |$32.38 \pm 2.25 \\%$|
> > | Meta-ticket | $71.57 \pm 0.31 \\%$ | $71.19 \pm 0.24 \\%$ | $71.12 \pm 0.34  \\%$  | $71.00 \pm 0.11  \\%$ | $70.90 \pm 0.32  \\%$  |
> >
> > As a reference, we also show the results of random masks (i.e., not meta-trained mask) applied to few-shot tasks.
> > Unfortunately, ProsPr performs very similarly to or even worse than random masks, whether with a single or multiple tasks.
> > This may be because ProsPr just computes the saliency scores, which are **enough information for large-scale training but too rough for few-shot learning**.
> > These results also indicate **the necessity of the meta-optimization as Meta-ticket does** particularly for few-shot learning.
> >
> > ## References
> >
> > [1] Alizadeh et al., "Prospect pruning: Finding trainable weights at initialization using meta-gradients." (ICLR 2022)
> >
> > [12] Lee et al. "Meta-learning with differentiable convex optimization." (CVPR 2019)
> >
> > [20] Tseng et al. "Cross-domain few-shot classification via learned feature-wise transformation." (ICLR 2020)

---

### Official Review · Reviewer_jF2G · 2022-07-13

**Rating:** 5
**Confidence:** 5
**Soundness:** 2 fair
**Presentation:** 2 fair
**Contribution:** 2 fair

**Summary:**

The authors proposed a novel meta-learning approach referred to as Meta-ticket, to find optimal sparse subnetworks for few-shot learning within randomly initialized NNs.

They empirically validated that Meta-ticket successfully finds subnetworks that can learn specialized features for each given task.

Due to this rapid task-wise adaptation ability, Meta-ticket achieves superior meta generalization compared to MAML-based methods, especially with large NNs.


**Questions:**

The Authors stated that Meta-ticket prefers rapid learning rather than feature reuse. On similar cross-domain meta-tasks, feature reuse might be preferred over rapid learning.

Is there any case that feature reuse is better than rapid learning in the Meta-ticket (adaptations such as specific to general or specific to specific)? A plot like Fig.2 would be helpful for better analysis of other adaptations.

The effects of the initial sparsity in B.2 (Fig.1 and Table 6) were a good analysis. The layerwise threshold hyperparameter seems to be sensitive to the initialization of weight scores. The authors could prepare the sensitivity analysis on various kinds of initialization methods with their iteration-validation performances.



**Limitations:**

The authors showed the limitation of subnetworks; Meta-ticket tends to degrade with small NNs compared to MAML.

To the issue, this work has found the solution to the degradation caused by the small NNs, providing iterative randomization techniques.


**Strengths And Weaknesses:**

The authors proposed a new Meta-ticket for few-shot learning, inspired by Lottery Ticket Hypothesis.

The Meta-ticket finds optimal sparse subnetworks which tend to adapt to each task with a large change in the overall parameters.

(+) The Meta-ticket is technically sound and seems to be supported by several experimental results.

(+) For fair comparisons with others, the authors prepared the cross-domain experimental settings and baselines – MAML, ANIL, and BOIL.

(-) The authors only showed the parts of adaptations such as general to specific and general to general.

(-) There is no ablation study on the layerwise threshold hyperparameter which seems to be sensitive to the initialization of weight scores.

---

> ### Author Response · Authors · 2022-08-01
> **Thank you for your review (1/2)**
>
> Thanks for your valuable review and suggestions.
>
> ### Results on specific to general/specific adaptations
>
> > (-) The authors only showed the parts of adaptations such as general to specific and general to general.
>
> In Section 4.1.3, we **actually showed the results** on the other types of adaptation such as specific to general and specific to specific.
> Please see Table 1 which contains the results on specific to general (VGG-Flower -> CIFAR-FS) and specific to specific (VGG-Flower -> Aircraft).
>
> Even so, thanks to your comment, we realized that such experiments with a larger CNN are missing.
> Thus we additionally conducted experiments for a larger CNN (ResNet-12) across more complex datasets (CUB to Imagenet or Stanford Cars).
> The following results show that the meta-generalization ability of Meta-ticket is effective even for the adaptation of specific to general/specific.
>
> | Methods | CUB -> CUB | CUB -> miniImagenet | CUB -> Stanford Cars |
> | :--- | :---: | :---: | :---: |
> | MAML | $78.92 \pm 0.62 \\%$ | $43.03 \pm 0.26 \\%$ | $38.95 \pm 0.42 \\%$ |
> | BOIL | $\mathbf{83.70 \pm 0.40} \\%$ | $49.17 \pm 1.30 \\%$ | $43.93 \pm 1.39 \\%$ |
> | Meta-ticket (Ours) | $80.49 \pm 0.50 \\%$ | $46.01 \pm 0.55 \\%$ | $40.24 \pm 0.92 \\%$ |
> | Meta-ticket + BOIL (Ours) | $83.28 \pm 0.44 \\%$ | $\mathbf{53.82 \pm 0.92} \\%$ | $\mathbf{48.85 \pm 0.56} \\%$ |
>
> From the results, we can see that even though BOIL (MAML-based method) achieves slightly higher accuracy than Meta-ticket + BOIL on the meta-training dataset (CUB) itself, however, Meta-ticket + BOIL **largely outperforms** BOIL evaluated on the other datasets.
>
> In conlusion, these results (Section 4.1.3 and the above additional experiments) reveal the effective meta-generalization ability of sparse NN structures **even for specific to general/specific adaptations**.
>
> ### Ablation study on layer-wise thresholds
>
> > (-) There is no ablation study on the layerwise threshold hyperparameter which seems to be sensitive to the initialization of weight scores.
>
> Sorry for confusing you.
> In fact, **the analysis in Appendix B.2** can be seen as an ablation study on the layer-wise score thresholds.
> This is because the layer-wise score thresholds are computed from the initial sparsity hyperparameter $p_{init}$ as described in Section 3.1.
> More precisely, for each $l$-th layer, the layer-wise score threshold $\sigma_l$ is computed as follows:
> Let $N_l$ be the number of parameters and $\\{s_i: 1\leq i \leq N_l\\}$ be the score parameters in the $l$-th layer.
> Then we set the threshold $\sigma_l$ as the $\lfloor p_{init} N_l \rfloor$-th smallest score $s_{i^*}$.
> By computing in this way, we can verify the ratio of $\\{s_i : s_i \leq \sigma_l\\}$ is equal to $p_{init}$, which is the current definition of the thresholds in Section 3.1.
>
> Nevertheless, thanks to your comment, we realized that the current definition may confuse readers.
> We will make it more precise in the final version of the paper as the above discussion.
>
> ### Q: Rapid learning vs feature reuse in specific to general/specific adaptations
>
> > The Authors stated that Meta-ticket prefers rapid learning rather than feature reuse. On similar cross-domain meta-tasks, feature reuse might be preferred over rapid learning. Is there any case that feature reuse is better than rapid learning in the Meta-ticket (adaptations such as specific to general or specific to specific)? A plot like Fig.2 would be helpful for better analysis of other adaptations.
>
> To confirm whether Meta-ticket prefers rapid learning even when meta-trained on a specialized dataset such as VGG-Flower, we conducted additional analysis on VGG-Flower like Fig. 2 in Section 3.2.
> In Figure 4 in the revised version of Appendix, we plotted and compared the dynamics of inner gradients between MAML and Meta-ticket during meta-training on VGG-Flower.
>
> Overall, similarly to the case of CIFAR-FS, Meta-ticket seems to prefer rapid-learning and MAML prefer feature reuse even on specialized datasets like VGG-Flower.
> Therefore, combining with the results on the specific to general/specific adaptations (given in the first section of this response), **the rapid-learning nature seems to be better also in the specific to general/specific cases** in our experiments.

---

> > ### Author Response · Authors · 2022-08-01
> > **Thank you for your review (2/2)**
> >
> > ### Q: Sensitivity of the hyperparameters and score initializations
> >
> > > The effects of the initial sparsity in B.2 (Fig.1 and Table 6) were a good analysis. The layerwise threshold hyperparameter seems to be sensitive to the initialization of weight scores. The authors could prepare the sensitivity analysis on various kinds of initialization methods with their iteration-validation performances.
> >
> > As discussed above, since the layer-wise thresholds are computed from the initial sparsity hyperparameter $p_{init}$, the analysis on the initial sparsity (Fig. 1 and Table 6 in Appendix B.2) also show the effects of the layer-wise thresholds.
> > To see the sensitivity of the initial sparsity more precisely, we also added the iteration-accuracy plots for several initial sparsities in Fig. 2 in the revised version of Appendix.
> > As we can see from Table 6 and Fig. 2 in the revised version of Appendix, both the final accuracy and training dynamics are **not so sensitive** to the initial sparsity hyperparameter (therefore, to the layer-wise thresholds too).
> >
> > Additionally, we conducted new experiments to see if the above tendency depends on the specific initialization method for score parameters.
> > Specifically, in this experiments, we initialize score parameters with Kaiming normal (KN) initialization and compare with Kaiming uniform (KU) initialization which was used in our paper.
> > The results on miniImagenet (5-shot 5-way) are as follows:
> >
> > | Initial Sparsity             | 0.0 | 0.3 | 0.5 | 0.7 | 0.9 |
> > | :---                         | :---: | :---: | :---: | :---: | :---: |
> > | Meta-ticket (KN init scores) | $71.95 \pm 0.53 \\%$  | $71.10 \pm 0.19 \\%$ | $71.12 \pm 0.29 \\%$  | $71.03 \pm 0.27 \\%$  | $70.96 \pm 0.25 \\%$ |
> > | Meta-ticket (KU init scores)  | $71.57 \pm 0.31 \\%$ | $71.19 \pm 0.24 \\%$ | $71.12 \pm 0.34  \\%$  | $71.00 \pm 0.11  \\%$ | $70.90 \pm 0.32  \\%$  |
> >
> > We can see that **the difference is very small between those two initialization methods** for scores.
> > Also, since the difference between the initial sparsities is less than $1 \\%$ in accuracy, the effect of the initial sparsity seems not so sensitive in this case too.

---

> > > ### Comment · Reviewer_jF2G · 2022-08-09
> > > **About Sensitivity of the hyperparameters and score initializations**
> > >
> > > Thank you for your rebuttal.
> > >
> > > I have checked the small sensitivities to the score initialization.

---

> > ### Comment · Reviewer_jF2G · 2022-08-09
> > **About additional experiments and explanations.**
> >
> > Thank you for the rebuttals - results on specific to general/specific adaptations, layer-wise thresholds, and rapid learning.
> >
> > In the results on specific general/specific adaptations, one possible analysis will be more helpful to better understand the meta-ticket, such as why was the meta-ticket not able to acquire a gain in CUB->CUB setting?

---

> > > ### Author Response · Authors · 2022-08-09
> > > **Thank you for your reply**
> > >
> > >
> > > > In the results on specific general/specific adaptations, one possible analysis will be more helpful to better understand the meta-ticket, such as why was the meta-ticket not able to acquire a gain in CUB->CUB setting?
> > >
> > > Thanks for posing such an interesting question.
> > > One possible reason is: MAML-based methods successfully encode useful features into the initial parameters to classify the fine-grained classes like bird species (on CUB), while Meta-ticket struggles to do so since their initial parameters are fixed with randomly initialized values.
> > > Also, as we observed in other experimental results (Table 1 and 2 in Section 4), the gain tends to be relatively small when evaluated on the meta-training dataset itself.
> > > Nevertheless, since it would be interesting if we could reveal such difference in nature between MAML and Meta-ticket, we are looking into conducting such an analysis (e.g., directly comparing the classification ability of the feature vectors before adaptation between MAML and Meta-ticket) for the final version of our paper.
> > > Thank you!

---

### Meta-Review · Area_Chair_VwoL · 2022-08-26

**Recommendation:** Accept
**Confidence:** Less certain

**Metareview:**

This paper proposes a meta-learning algorithm which aims to find a subnetwork of a randomly initialized network, that is optimized for each task. Specifically, the authors propose to meta-learn the latent continuous parameters that correspond to a task-specific binary mask for each task, and provide empirical evidence and theoretical analysis which show that the method does rapid learning for each task rather than reusing the meta-learned features. The proposed Meta-ticket algorithm is validated on the few-shot classification tasks from various domains, and is shown to outperform existing meta-learning algorithms such as MAML, ANIL, and BOIL.

The reviewers in general were positive about the idea of finding a lottery ticket, or an optimal subnetwork for each task in a meta-learning scenario, and the discussion on rapid learning v.s. feature reuse. Also, they found the experimental validation as sufficiently extensive and the paper well-written.

However, there was a major concern regarding the work’s limited novelty over existing works that aim to learn a binary masking in a meta-learning, or multi-task scenarios, such as [Alizadeh et al. 2022], [Mallya et al. 2018]. Also, while not mentioned by the reviewers, [Lee et al. 20] cited in the paper also aim to learn a task-adaptive masking of shared initialization parameters. While the authors provided experimental comparison to ProsPr [Alizadeh et al. 2022], other methods were not discussed or compared against despite their relevance. There were also other minor concerns, such as missing ablation study on the hyperparameter, degraded performance with small neural networks, but they were mostly addressed away in the author responses.

In sum, this is a well-written paper that provides a fresh view of the task-adaptation in the meta-learning in the perspective of lottery ticket hypothesis and feature reuse v.s. Rapid adaptation argument. However, due to limited novelty, discussion, and experimental comparison and analyses of the comparative study over existing works, I recommend a borderline accept for this paper in its current state. The authors are strongly advised to include discussion of [Mallya et al. 18], and experimental comparison against [Lee et al. 20] in the final version of the paper.

[Mallya et al.] Piggyback: Adapting a Single Network to Multiple Tasks by Learning to Mask Weights, CVPR 2018
[Lee et al. 20] Learning to Balance: Bayesian Meta-Learning for Imbalanced and Out-of-distribution Tasks, ICLR 2020
[Alizadeh et al. 22] Prospect Pruning: Finding Trainable Weights at Initialization using Meta-Gradients, ICLR 2022

**Award:**

No

---

### Decision · Program_Chairs · 2022-09-14

Accept